# Effect of Antimony Buffer Layer on the Electric and Magnetic Properties of 200 and 600 nm Thick Bismuth Films on Mica Substrate

**DOI:** 10.3390/ma13092010

**Published:** 2020-04-25

**Authors:** Elena S. Makarova, Anastasiia S. Tukmakova, Anna V. Novotelnova, Vladimir A. Komarov, Vasilisa A. Gerega, Natallya S. Kablukova, Mikhail K. Khodzitsky

**Affiliations:** 1Faculty of Cryogenic Engineering, ITMO University, Saint-Petersburg 197101, Russia; astukmakova@itmo.ru (A.S.T.); novotelnova@itmo.ru (A.V.N.); 2Faculty of Physics, Department of General and Experimental Physics, Herzen State Pedagogical University of Russia, Saint Petersburg 191186, Russia; va-komar@yandex.ru (V.A.K.); gerega.vasilisa96@gmail.com (V.A.G.); 3International Scientific and Research Institute of Bioengineering, ITMO University, Saint-Petersburg 197101, Russia; kablukova.natali@yandex.ru (N.S.K.); khodzitskiy@yandex.ru (M.K.K.); 4Saint Petersburg State University of Industrial Technologies and Design, Saint-Petersburg 191186, Russia; 5Terahertz Biomedicine Laboratory, ITMO University, Saint-Petersburg 197101, Russia

**Keywords:** thin film, bismuth, galvanomagnetic properties, resistance, relative magnetoresistance, hall coefficient

## Abstract

We report on the production of 200 and 600 nm thick Bi films on mica substrate with 10 nm thick Sb sublayer between Bi and mica. Two types of films have been studied: block and single crystal. Films were obtained using the thermal evaporation technique using continuous and discrete spraying. Discrete spraying allows smaller film blocks size: 2–6 μm compared to 10–30 μm, obtained by the continuous spraying. Single crystal films were made by the zone recrystallization method. Microscopic examination of Bi films with and without Sb sublayer did not reveal an essential distinction in crystal structure. A galvanomagnetic study shows that Sb sublayer results in the change of Bi films properties. Sb sublayer results in the increase of specific resistivity of block films and has no significant impact on single crystal films. For single-crystal films with Sb sublayer with a thickness of 200 nm the Hall coefficient has value 1.5 times higher than for the 600 nm thickness films at 77 K. The change of the Hall coefficient points to change of the contribution of carriers in the conductivity. This fact indicates a change in the energy band structure of the thin Bi film. The most significant impact of the Sb sublayer is on the magnetoresistance of single-crystal films at low temperatures. The increase of magnetoresistance points to the increase of mobility of the charge carriers. In case of detecting and sensing applications the increased carriers mobility can result in a faster device response time.

## 1. Introduction

Development of the devices, operating in terahertz (THz) frequency range, is one of the rapidly evolving areas of modern physics. The interest in the development of efficient THz devices is determined by the breadth of practical applications: the information transfer rate improvement, the increase in safety, for example in airports, replacing X-ray systems with THz scanners, in medicine and other fields [1,2,3,4]. Production technologies of various emitters and sensors of terahertz radiation are developed [5] on basis of narrow-band-gap semiconductors. The energy band gap affects the absorption capacity of THz emission in the materials. Thus, it is possible to expect a significant increase in THz absorption efficiency in materials with small overlap between energy bands compared to THz emission energy. The response of charge carriers will be faster in the materials with small overlap between energy bands and small effective mass of the carriers.

It was shown [6,7] that the most promising materials for THz devices may be semimetals: Bi, Sb, As, etc. All of them are solid solutions. The response of transport phenomena on THz emission in single-crystal Bi was discovered in the work [6]. Bismuth is anisotropic material, its properties are sensitive to crystal orientation [6]. The films of bismuth are sensitive to the classic size effect [8]. It was shown that the film thickness limits the mobility of electrons. THz emission interacts better with the p-type materials [9], i.e., with the materials with positive Seebeck and Hall coefficients.

The films are specified by a thickness that will limit the electrons, but the holes will have a mean free path as in a massive crystal. This fact will enhance the contribution of holes to the conductivity and will result in the positive Hall coefficient. In block films, the mean free path of charge carriers is limited by the thickness and size of the blocks. To enhance the effect, it is essential to remove the boundaries between the blocks, i.e., to get a single-crystal film. In [8] it was demonstrated that the ultrathin antimony layer can change the sign of Hall coefficient in the block films of Bi.

This research is the extension of the preceding our works [8,10]. Both papers consider the impact of antimony sublayer. In [8], the properties of Bi block films of different thickness have been presented. Another paper [10] focused on the impact of substrate material on the films galvanomagnetic properties.

The current work presents a more extensive study of Bi films with different thickness (200 and 600 nm) and with different structure (block and single crystal). It is focused on the study of the effect of Sb sublayer on the electric charge transfer phenomenon in Bi thin films. In addition, the investigation of deposition method impact on the films structure has been carried out. The results of measurements of resistivity, magnetoresistance and Hall coefficient were performed. The properties of the films are affected by the classical size effect, so different crystalline structures of the films were chosen: block and single-crystal.

## 2. Materials and Methods

### 2.1. Technology of Production of Thin Bi Films and Sb Ultralayer

Thermal spraying method was used to obtain thin films. This method consists of transferring the substance into the gas phase by evaporation and condensing the substance onto the substrate which temperature is lower than gas temperature. There is a vacuum at pressure of $10^{-5}$ Torr in the setup chamber. The evaporation temperature at operation pressure is 600 ${}^{\circ }$C for Bi and 700 ${}^{\circ }$C for Sb. In this research two types of thermal spraying method were used: discrete and continuous. Within the continuous thermal spraying method the required substance mass is placed on the evaporator. Within the discrete method the substance is placed into the evaporator in portions. For both methods the composition of film remained constant. Hence, the difference between discrete and continuous methods is only related to the method of substance delivery to the evaporator. Deposition mode, spraying rate, substrate temperature during thin film formation, annealing time and temperature are the factors which affect the structure of thin films [11,12,13].

The block films were produced on the substrate with the temperatures of 120 ${}^{\circ }$C for Bi, 150 ${}^{\circ }$C for Sb and at the annealing temperature of 250 ${}^{\circ }$C within 30 min (Table 1). Bi and Sb have a purity of 99.9%.

The single-crystal thin films were made by the zone recrystallization method under the coating [14] of blanks produced by thermal spraying method. The antimony has the higher melting point than bismuth (the melting point is 271.4 ${}^{\circ }$C for Bi and 630.5 ${}^{\circ }$C for Sb), so the sublayer of antimony within the zone recrystallization treatment stays in solid state. The blanks for single-crystal films were produced at the substrate temperature of 20 ${}^{\circ }$C for Bi, 150 ${}^{\circ }$C for Sb, the annealing was not done (Table 1).

Two types of structures were obtained (Figure 1): bismuth films with 10 nm thick antimony sublayer, and for comparison, films obtained in the same technological modes, but without a sublayer.

### 2.2. Method of Films Thickness Measurement

The dependence of film thickness on the amount of the loaded substance in used in thermal spraying method, and makes it possible to set the film thickness before spraying. To obtain the films with a certain thickness the method of metered weighing was used. The exact thickness of the films were measured by using Linniks interferometer MII-4. The method of the film thickness measurement is based on the observation of two systems of interference bands displaced relative to each other [15,16,17]. An interference pattern is formed by the interaction of light beams reflected from the surface of a substrate partially covered by the layer under study and a reference mirror.

The thickness of the ultrathin antimony layer was measured using an atomic force microscope using the step method [18].

### 2.3. Structure Research Methods

#### 2.3.1. Methods of Studying the Surface and Structure of Block Films

Atomic force microscopy (AFM) is a modern method that allows exploring the surface of the object under study at the nanoscale. The film structure was studied in air using a Solver P47PRO scanning probe microscope (NT-MDT) by atomic force microscopy (AFM) in the semicontact mode. Cantilevers have a resonance frequency of about 150 kHz [19,20]. For clear identification of crystallites boundaries of thin films, chemical etching was carried out in a solution of nitric and acetic acids. This solution makes it possible to reveal crystallite boundaries and dislocation etch pits. During the study of crystallites boundaries by an atomic force microscope, the difficulty arises due to the fact that it can examine a limited part of the surface, while the dimensions of the film are much larger. Therefore, to study the films after etching, another optical method was applied. In this work, a Nikon Eclipse LV 150 optical microscope was used. This method allows a quick examination of the entire film surface and analysis of the crystallites size in different films.

#### 2.3.2. X-ray Diffraction

Investigation of the structure and composition of the obtained films were carried out using an X-ray diffractometer. The crystal rotation method was used. This method includes the rotation of a crystal around a certain crystallographic orientation and the irradiation of this crystal with monochromatic characteristic X-ray radiation [21,22]. The diffraction pattern is recorded, analyzed by a special program and displayed on a computer monitor. Diffraction patterns were obtained using a Phywe XR equipped with a Cu $K_{\alpha }$ source [23].

### 2.4. Method of Galvanomagnetic Properties Measurement

To study the transport phenomena, the galvanomagnetic properties of the films were measured using an automated installation, which has a range of temperature regimes from 77 to 310 K with a magnetic field induction of 0.65 T. The measurements were carried out at a constant temperature and a constant magnetic field, with a given temperature gradient. The 4-probe method was used. The installation performed measurement at 17 temperature points and at 6 fixed values of the magnetic field induction. Thin film was placed in a special holder. The temperature of the sample was measured by two copper-constantan thermocouples mounted on the edges of the copper blank. The accuracy of temperature measurement was 1 K. In the working area of holder, a vacuum of about 5 Pa was maintained during the entire measurement. The resistivity, the relative magnetoresistance, and the Hall coefficient were measured within the galvanomagnetic properties studying. The measurement errors were 6% for the resistivity, 4% for the magnetoresistance, and 10% for the Hall coefficient.

## 3. Results and Discussion

### 3.1. Thin Films Structure

#### 3.1.1. X-ray Diffraction Study

The diffraction patterns of the films are presented in Figure 2a (the red line), there are the peaks that not only belong to the bulk Bi(111) (Figure 2b, the black line), but also to the substrate in Figure 2a (the black line). To determine the structure of the obtained film, it is necessary to determine the peaks that belong to the film of Bi itself. In the diffraction patterns, a good resolution of the 5th order maximum (at $2\Theta =152,9^{\circ }$) for $K_{\alpha }$ copper lines is observed, indicating a good crystalline structure of the sample.

The peak shape for the films follows the peak shape for the crystal of the corresponding orientation (Figure 2b), $C_{3}$ perpendicular to the plane of the film or Bi(111). The 5th order maximum is well resolved and does not shift toward large angles (Figure 2b), hence, antimony did not dissolve in the volume of bismuth. At low Sb concentrations in the bismuth-antimony solid solution, the crystal lattice parameter changes, and the 5th order maximum shifts toward large angles (example, $Bi_{97}Sb_{3}$$2\Theta =153,6^{\circ }$).

#### 3.1.2. Ultra Thin Sb Layer Structure

To identify the effect of the ultrathin antimony sublayer on the structure of the bismuth film, the Sb layer was separately studied using the atomic force microscope. The study of ultrathin antimony film with the thickness of 10 nm showed that there were no orienting crystal shapes on the surface of the film (Figure 3). Ultrathin antimony film cannot significantly influence the formation of Bi film structure from vapor phase [8].

#### 3.1.3. Block Bismuth Films Structure

AFM–study of the surface of annealed bismuth films showed that the crystal growth figures have a triangular shape (Figure 4). This shape of the crystal growth figures indicates that the crystallographic axis $C_{3}$ is directed perpendicular to the plane of the film. By the nature of the mutual arrangement of the growth figures, it can be argued that the film has a block structure. The border of the block is considered to be the boundary between the triangles with the opposite orientation (the upper right corner of Figure 4a). The blocks have a size of 2–5 micrometers (Figure 4).

Chemical etching makes it possible to see the boundaries of crystallites more clearly than it is shown in Figure 4. After the etching, when the boundaries are destroyed, the crystallites (blocks) can be easily seen in the AFM–image. The bismuth films produced by continuous thermal spraying have crystallites of several tens of micrometers long (Figure 5a). The bismuth films produced by discrete spraying method have the size of crystallites 2–6 micrometers (Figure 5b).

This study showed that the evaporation method has a greater effect on Bi thin films crystallite size than a thin sublayer of antimony.

#### 3.1.4. Single-Crystal Bismuth Films Structure

Since the dimensions of the film are larger than the maximum possible scanning area provided by atomic force microscopy, the main analysis was performed using an optical microscope. In addition, AFM examination of the surface of bismuth single-crystal film does not provide information on the structure of the obtaining film, because the film grows in the direction of recrystallization zone parallel to the substrate plane. Chemical etching and X-ray diffraction analysis were used for structure analysis. After chemical etching of zone recrystallized bismuth films (Figure 6), no crystallite boundaries appear. Dislocation etching pits have triangular shape, and all etching pits have the same orientation on the surface of the entire film. This indicates that the recrystallized bismuth film is a monocrystalline.

### 3.2. The Study of the Galvanomagnetic Properties

A study of ultrathin antimony layer of 10 nm thick showed that its resistivity 15 times greater than for Bi films. The size effect on resistivity is observed for all samples: as thickness decreases, specific resistivity increases (Figure 7). The sublayer affects in different ways on the values of transport coefficients. For 200 nm single-crystal films antimony sublayer reduces the resistance. For 600 nm films antimony sublayer can be considered to have no impact in the investigated temperature range. In the work [8] the effect of reduced resistivity in block films with the thickness of 650 and 90 nm was observed to be similar for single-crystal films.

The Sb sublayer in single-crystal films increases the value of the relative magnetoresistance by 1.5–2 times at 77 K. The more magnetoresistance the higher the mobility of charge carriers in films (Figure 8). An increase in film thickness, block size (up to a single crystal), and the presence of the Sb sublayer increase the magnetoresistance. The effect of the Sb sublayer on the magnetoresistance decreases with increasing bismuth film thickness.

The Hall coefficient for almost all the thin films has positive values (Figure 9). In block bismuth films, the Hall coefficient at low temperatures in some cases can also take negative values, and with increasing crystallite sizes, the Hall coefficient tends to positive values. The Hall coefficient for single-crystal films has positive values over the entire temperature range and increases with temperature decrease. For films with a thickness of 200 nm the Hall coefficient has the higher value, than for films with a thickness of 600 nm. This is possibly due to a consequence of the characteristics of the interaction of charge carriers with the surface of a bismuth film. A positive value of the Hall coefficient for single-crystal films of any thickness with an orientation of the $C_{3}$ axis of the film perpendicular to the substrate indicates that a negative value of the Hall coefficient in films obtained by thermal evaporation is associated with their block structure.

## 4. Conclusions

The crystal structure of block bismuth films as part of the layered structure showed that ultrathin antimony sublayer (10 nm) had no significant effect on bismuth films structure. For films both with and without Sb sublayer the $C_{3}$ axis was normal to the film plane, films had a block structure and crystal growth figures had a triangular shape.

The size of the crystallites in block bismuth thin films is greatly affected by the deposition regime. The crystallites of the films obtained by the continuous spraying have larger size than that obtained by the discrete spraying; the size is approximately 2–3 times larger.

On the other hand, the study of galvanomagnetic properties showed that the Sb sublayer impacts on values of the Hall coefficient, the magnetoresistance and the specific resistivity. With a decreasing thickness of the films with Sb sublayer, an increase of the positive Hall coefficient value is observed. The positive Hall coefficient proves the enhance of the holes contribution to the conductivity. The temperature dependence of the magnetoresistance showed that the Sb sublayer increases the mobility of the charge carriers in single-crystal films. The increased mobility of the charge carriers and the change in the contribution to conductivity indicate a change in the energy band structure of the thin film of Bi.

The increased charge carriers mobility can result in smaller response time of detectors and sensors based on Bi films. Hence, the use of Sb sublayer can help to improve both mechanical reliability and efficiency of the devices based on Bi thin films.

## Figures and Tables

**Figure 1 materials-13-02010-f001:**
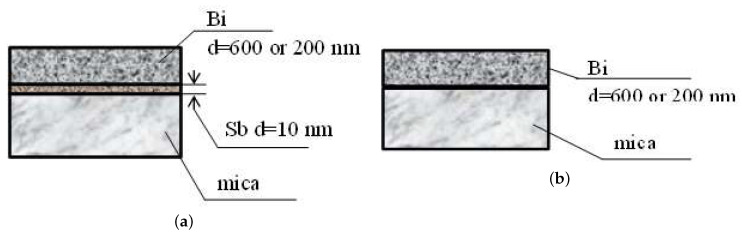
Schematic representation of the structures under study: (**a**) without antimony sublayer (**b**) with 10 nm antimony sublayer.

**Figure 2 materials-13-02010-f002:**
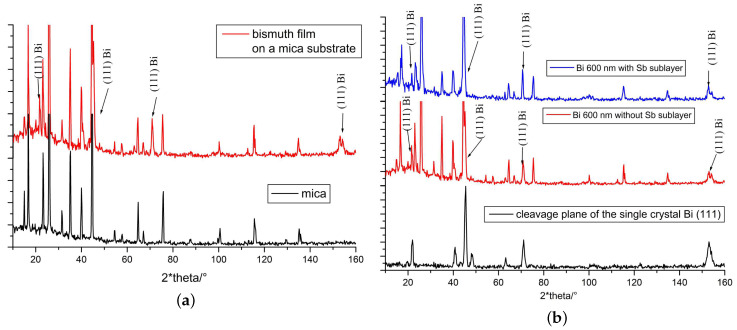
Diffractograms of thin films: (**a**) mica and bismuth films on mica, (**b**) cleavage planes of bismuth single crystals of orientation (111), and bismuth films with a base and without an antimony sublayer.

**Figure 3 materials-13-02010-f003:**
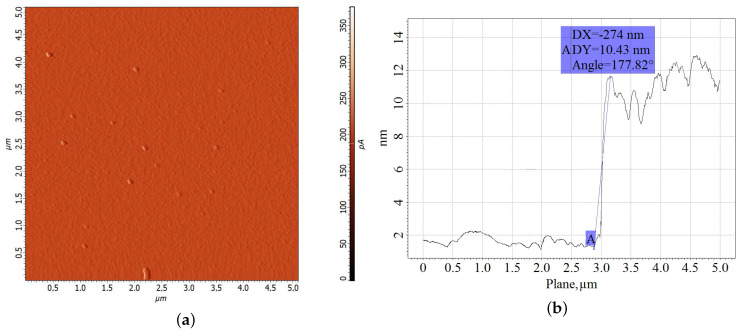
AFM–image: (**a**) of Sb film surface with 10 nm thickness deposited on mica, (**b**) measurement of ultrathin antimony layer thickness using the AFM method.

**Figure 4 materials-13-02010-f004:**
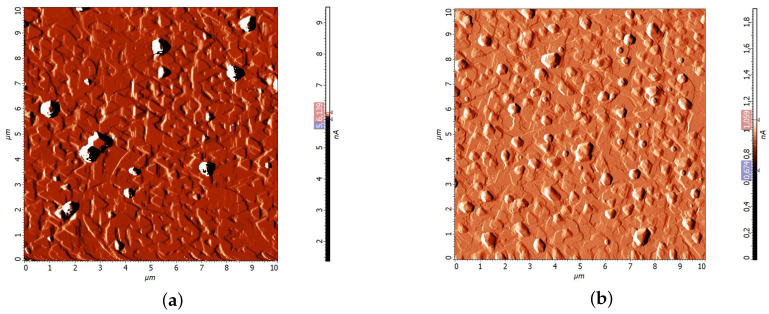
AFM–image of Bi film surface: (**a**) without antimony sublayer (**b**) with antimony sublayer. Films have been obtained by the discrete spraying method.

**Figure 5 materials-13-02010-f005:**
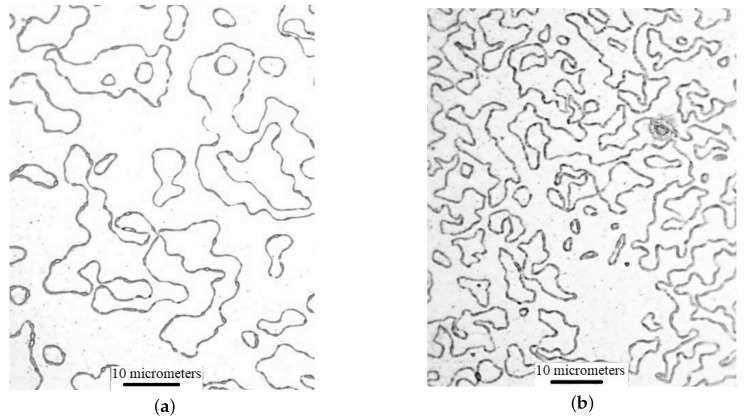
Photograph of etched bismuth film surface on mica using optical microscope Nikon Eclipse LV 150: (**a**) continuous thermal spraying (**b**) discrete spraying method.

**Figure 6 materials-13-02010-f006:**
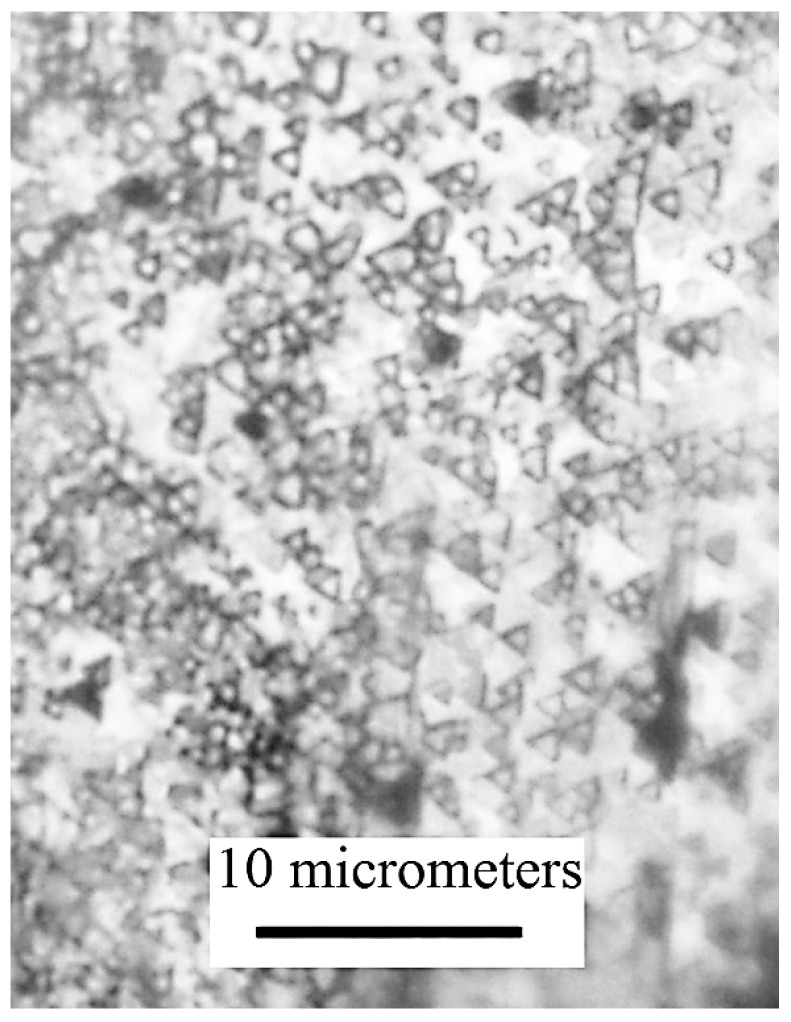
Photograph of the etched surface of a monocrystalline bismuth film on mica made using a Nikon Eclipse LV 150 optical microscope.

**Figure 7 materials-13-02010-f007:**
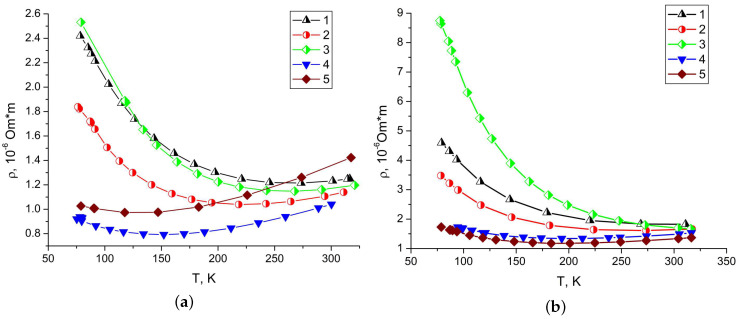
Temperature dependence of bismuth films specific resistivity (**a**) 600 nm thick and (**b**) 200 nm thick. Block bismuth films without Sb sublayer sprayed with discrete (1) and continuous (2) evaporation method; block film with Sb sublayer (3); single-crystal films without Sb sublayer (4) and with Sb sublayer (5).

**Figure 8 materials-13-02010-f008:**
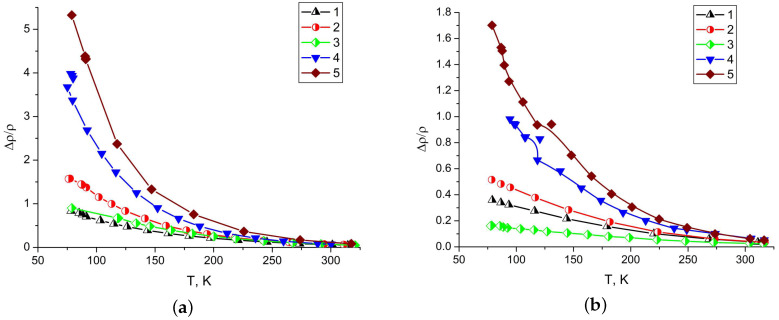
Temperature dependence of the relative magnetoresistance of bismuth films (**a**) 600 nm thick and (**b**) 200 nm thick. Block bismuth films without Sb sublayer sprayed with discrete (1) and continuous (2) evaporation method; block film with Sb sublayer (3); monocrystalline films without Sb sublayer (4) and with Sb sublayer (5).

**Figure 9 materials-13-02010-f009:**
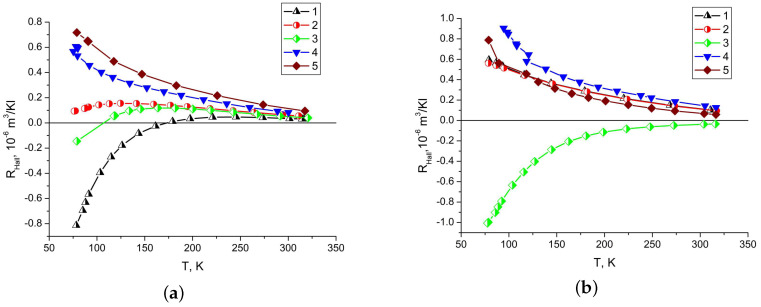
Temperature dependence of Hall coefficient of bismuth films (**a**) 600 nm thick and (**b**) 200 nm thick. Block bismuth films without Sb sublayer sprayed with discrete (1) and continuous (2) evaporation method; block film with Sb sublayer (3); monocrystalline films without Sb sublayer (4) and with Sb sublayer (5).

**Table 1 materials-13-02010-t001:** Technological modes of thin films production.

Films	Number of Layers	Substrate Temperature, ${}^{\circ }$C for Sb Ultrathin Layer	Substrate Temperature, ${}^{\circ }$C for Bi Film	Annealing Temperature, ${}^{\circ }$C
Blank for zone recrystallization	1	-	20	-
	2	150	20	-
Block films with crystallite sizes greater than film thickness	1	-	120	250
	2	150	120	250

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
