# Peer review of "Effect of Antimony Buffer Layer on the Electric and Magnetic Properties of 200 and 600 nm Thick Bismuth Films on Mica Substrate"

_materials, 2020, doi:10.3390/ma13092010_

Round 1

Reviewer 1 Report

This manuscript reports an investigation into the effect of an antimony interlayer on the structure and transport properties of bismuth thin films deposited by thermal spraying. The research appears to have been undertaken in a professional manner and, as far as I am aware, is reasonably novel. The figures have been properly presented and there is adequate reference to the literature.

There are some structural issues with the manuscript that ought to be addressed.

The standard of the English, while being reasonably easy to understand, is at the borderline of that that is publishable; I would recommend taking some advice from a fluent speaker of English.

The abstract lacks some detail, in particular the method of deposition and the substrate are note mentioned there. The abstract should be re-written to summarise the research work, including the main outcomes. As it stands it is a description of the paper rather than a precis.

I could not find reference to Figure 1 in the text.

Section 5 is entitled Results and Discussion, which I find a little odd as the results have already been presented in sections 3 and 4. The discussion is also sparse. I would like to see some more discussion of the underlying science. For example is the large Fermi wavelength of bismuth a factor in the size effects? How do the observed structural and transport properties interrelate and why? Please expand the Discussion.

What are the error bars on Figures 6, 7 and 8?

Reviewer 2 Report

"Properties of bismuth thin films on the antimony sublayer"

The Authors present and discuss results on the possibility of using antimony ultrathin layer as intermediate/sublayer for improving the adhesion between bismuth film and substrate. Several aspects need careful and thorough revision, please see the comments below; the manuscript should not be accepted for publication in its present form.

* Title: not meaningful; please use something more related to your actual work;
* Highlights & Graphical abstract: not provided;

Abstract
* not meaningful, please rephrase it and insist on the novelty and importance of your work, provide more and actual data;
* use present tense (recommended), avoid speculative and /or unnecessary words (e.g. ultrathin, significant).

1. Introduction
* this section is incongruent, presenting lots of aspects, superfluous; please rephrase it, be concise on the context and please do a thorough search about this topic;
* insist more on the novelty and importance of your work, with respect to literature; further discuss with respect to your previous work (https://doi.org/10.1134/S1063782617070168, https://doi.org/10.1088/1742-6596/1400/5/055048);
* the last paragraph of this section should be a short description of your work.

2. Experimental (Methodology)
* thermal spraying: please provide further details and adequate references;
* sample preparation: please further present the preparation method and provide more references to it, if possible;
* provide in detail all analysis techniques and equipment;
* antimony thin film: further explain on how the thickness and roughness is measured;
* all results should be presented and discussed in the next section, not here.

3. Results and discussion
* as a general overview / remark to this section: please include here all results and discuss them in a correlated manner;
* please start the section by describing the main aspects of your work - what do you seek and what is your plan in doing so (a few phrases will suffice); what is the novelty with respect to your previous work and literature?
* Figure 2: please further discuss the image in text and your choice of substrate;
* Figure 3: further discuss the images in text (one may observe a certain crystallographic structure)
* Figures 4 & 5: all details of the experimental techniques and equipment should be presented in section 2 (Experimental);
* Figure 6, 7 & 8: define the numbers (1-5) in the graphs;
* section "5. Results and discussion" should be included here.

4. Conclusion
* this section is not meaningful, please rephrase it to emphasize more on the novelty and importance of your work with respect to literature;
* further provide data on the most important aspects (results) of this study.

Minor aspects
* references: this section is missing important publications;
* avoid providing redundant data, superfluous text, or the repetitive use of speculative words or phrases;
* check the manuscript for typographical errors (language, spelling, punctuation, numbering of sections, etc).

Reviewer 3 Report

Dear Authors,

in my opinion, your manuscript titled: "Properties of bismuth thin films on the antimony sublayer" can be published in Catalysts after major revision.

Please, you can find remarks, suggestions and proposals of corrections below.

In my opinion, you have to describe, what is the main novelty in your work because you refer to earlier studies and it is difficult to say what is the main goal of your work.

In my opinion, you have to perform the characterisation of materials using atomic force microscope (AFM) and transmission or scanning electron microscopy (TEM or SEM) after the studies of galvanomagnetic properties.

You have to show the results of X-ray diffraction analysis obtained for materials, before and after their application in the studies of galvanomagnetic properties.

Please, can you show the results which evidence the real thickness of the sublayer of antimony and the layer of bismuth in the studied films.

You have to record the STEM images combined with EDX spectra and you have to apply the XPS technique for the studies of films because these measurements can be applied to estimate the purity of antimony and bismuth layers before and after the application of films in the studies of galvanomagnetic properties.

In my opinion, you have to add to the manuscript the information about the procedure of materials characterisation using atomic force microscope (AFM) and optical microscope. You have to add also the information about the procedure of galvanomagnetic tests.

I think that you have to change the style of manuscript text because you use many repetitions of words in consecutive sentences, e.g.

The dependence of Hall coefficient on temperature is shown in Figure 8. The Hall coefficient for recrystallized films has positive values over the entire temperature range and increases with temperature decrease. The increase of Hall coefficient with the reduction of film thickness is also observed for zone recrystallized films.

Kind regards,
Reviewer 

Round 2

Reviewer 2 Report

"Properties of bismuth thin films on the antimony sublayer" - R1

New title: "Study of electrics characteristics of monocrystal thin films of bismuth without antimony sublayer on mica"

The Authors incompletely and/or incorrectly addressed the issues raised during the peer-review process. Most of the aspects still need thorough revision. Furthermore, I must insist that the title, the abstract and the sections of the manuscript to be written in a more correlated manner.

Title: still not meaningful, please use something more related to your actual work, to better emphasize your work; for example, use something such as: "Effect of antimony buffer layer on the growth and properties of bismuth thin films". 

Abstract: not meaningful, please rephrase it and insist on the novelty and importance of your work, provide more and actual data; use present tense; avoid speculative and / or unnecessary words.

1. Introduction: please rephrase it, be concise on the context and please do a thorough search about this topic; insist more on the novelty and importance of your work, with respect to literature and in particular to your previous work (some similarities are too evident / autoplagiarism with, for example: https://doi.org/10.1134/S1063782617070168, https://doi.org/10.1088/1742-6596/1400/5/055048).

2. Experimental: please rephrase this section to make it more coherent.

3. Results and discussion: as a general overview / remark to this section, the results need to be discussed in a more correlated manner.
* Figure 2: please further discuss the images in text, and index the peaks; what is one supposed to see here, what do each of the peaks stand for? 
* Figure 3: provide the z-axis; further explain in text;
* Figure 4: remove it, just make a discussion in text; 
* Figure 5: please further discuss the images in text and provide the z-axis;
* Figures 6 and 7: please further discuss the images in text;

4. Conclusion: please rephrase this section to emphasize more on the novelty and importance of your work with respect to literature, and further provide data that underline the most important aspects (results) of your study.

Minor aspects: avoid providing redundant data, superfluous text, or the repetitive use of speculative words or phrases; please thoroughly correct the manuscript for typographical errors - language, spelling, punctuation, numbering of sections, etc.

The manuscript should not be accepted for publication in its present form.

Reviewer 3 Report

ear Authors,

in my opinion, your manuscript titled: "Study of electrics characteristics of monocrystal thin films of bismuth without antimony sublayer on mica" can be published in Catalysts after major revision.

In my opinion, you have to perform the characterisation of materials using atomic force microscope (AFM) and optical microscope after their  galvanomagnetic tests. You have to compare the properties of materials before and after their performance in these tests. In my opinion, you have to record the TEM or SEM images of materials before and after their application in the studies of galvanomagnetic properties, because the images obtained using optical microscope don't show many details in the morphology of films.  

Kind regards,
Reviewer 

Round 3

Reviewer 3 Report

Dear Authors,

in my opinion, your corrected manuscript titled: "Effect of antimony buffer layer on the electric and magnetic properties of 200 and 600 nm thick bismuth films on mica substrate" can be published in Catalysts after minor revision.

You have to add the results of the materials characterisation using selected analytical techniques, e.g. atomic force microscope (AFM), after the galvanomagnetic tests. The conclusions of research should be supported by the results of materials characterisations after the studies of galvanomagnetic properties. 

You have to perform the second cycle of galvanomagnetic tests (the reusing of samples). You have to compare the obtained results with the results from the first cycle. This comparison can show the possible stability of tested materials. 

Kind regards,
Reviewer 
